# What outcomes are associated with developing and implementing co-produced interventions in acute healthcare settings? A rapid evidence synthesis

David Clarke,[1] Fiona Jones,[2] Ruth Harris,[3] Glenn Robert,[3]  on behalf of the Collaborative Rehabilitation Environments in Acute Stroke (CREATE) team

[1]Academic Unit of Elderly Care and Rehabilitation, Leeds Institute of Health Sciences, Bradford, UK
[2]Faculty of Health Social Care and Education, St George's University of London, London, UK
[3]Florence Nightingale Faculty of Nursing and Midwifery, King's College London, London, UK

**Correspondence to**
David Clarke; d.j.clarke@leeds.ac.uk

## ABSTRACT

**Background** Co-production is defined as the voluntary or involuntary involvement of users in the design, management, delivery and/or evaluation of services. Interest in co-production as an intervention for improving healthcare quality is increasing. In the acute healthcare context, co-production is promoted as harnessing the knowledge of patients, carers and staff to make changes about which they care most. However, little is known regarding the impact of co-production on patient, staff or organisational outcomes in these settings.
**Aims** To identify and appraise reported outcomes of co-production as an intervention to improve quality of services in acute healthcare settings.
**Design** Rapid evidence synthesis.
**Data sources** Medline, Cinahl, Web of Science, Embase, HMIC, Cochrane Database of Systematic Reviews, SCIE, Proquest Dissertation and Theses, EThOS, OpenGrey; *CoDesign*; *The Design Journal*; *Design Issues*.
**Study selection** Studies reporting patient, staff or organisational outcomes associated with using co-production in an acute healthcare setting.
**Findings** 712 titles and abstracts were screened; 24 papers underwent full-text review, and 11 papers were included in the evidence synthesis. One study was a feasibility randomised controlled trial, three were process evaluations and seven used descriptive qualitative approaches. Reported outcomes related to (a) the value of patient and staff involvement in co-production processes; (b) the generation of ideas for changes to processes, practices and clinical environments; and (c) tangible service changes and impacts on patient experiences. Only one study included cost analysis; none reported an economic evaluation. No studies assessed the sustainability of any changes made.
**Conclusions** Despite increasing interest in and advocacy for co-production, there is a lack of rigorous evaluation in acute healthcare settings. Future studies should evaluate clinical and service outcomes as well as the cost-effectiveness of co-production relative to other forms of quality improvement. Potentially broader impacts on the values and behaviours of participants should also be considered.

## Strengths and limitations of this study

► This study is the first to systematically review outcomes associated with developing and implementing co-produced interventions in acute healthcare settings.
► We have identified a lack of rigorous evaluation of effectiveness and cost-effectiveness of co-produced interventions in the acute healthcare sector at both the service and system levels.
► The study did not access and review the much broader patient and public involvement, service improvement and clinical microsystem literatures unless specific terms such as 'co-production' and 'co-design' were used.
► We only included studies where full-text reports published in English within the last 10 years could be retrieved; co-production studies published in other languages may have been overlooked by our review.

## BACKGROUND

There is renewed interest in and advocacy for adoption of 'co-production' as a means of co-creating value across the public sector. However, as the term increasingly enters mainstream management (and healthcare quality improvement) discourses, there is a sense that it may be being misused or misunderstood, thereby running the risk of becoming denuded of meaning or losing any association with its radical roots.

Co-production was first conceptualised in the USA in the 1970s, originally as a response to the lack of recognition of service users in service delivery.[1] The creation of time banks, a system reliant on the participation of volunteers who are also service users,[2 3] showed how collaborative interventions that involve people with long-term psychosocial needs could contribute to improved community

links.[4] Since this early work, a variety of terms have become evident in the growing literature on co-production, and a range of related approaches are being enacted in different ways and at various levels throughout public and health sector services.[5–11] For the purposes of this paper, we follow Osborne *et al*[8] recent definition of co-production as 'the voluntary or involuntary involvement of public service users in any of the design, management, delivery and/or evaluation of public services'.

In the healthcare context, co-production is promoted as harnessing the knowledge of patients, carers and staff to make changes about which they care most.[12 13] A recent example of using co-production in this way is a project to deliver and evaluate a mental health service to meet the needs of black and minority ethnic communities.[14] Berwick[15] has recently argued for a greater focus on such co-production as part of a proposed 'third era' of medicine. Potential benefits might include from a patient perspective improved satisfaction with the experience of care, reduction in complications associated with treatment and improved health outcomes.[12–14] From a staff and organisational perspective, the case for co-production may include improvements in staff well-being, increased recognition of the need for a better understanding of patient perspectives and changes in attitudes toward working with patients as partners in quality improvement.[12–15]

In a recent commentary, Batalden *et al*,[12] while arguing that where healthcare activities are co-produced, services, providers and service users become far more effective agents of change, note that current systems can both support and constrain partnerships between patients and professionals (and that historically this kind of partnership has been unequal). The evidence base in terms of robust evaluation of the impact of co-production on patient outcomes appeared limited. Batalden *et al*[12] argue that this raises a series of questions for proponents of co-production including the need to determine '[m]easures that accurately reflect co-produced service and its results [and] measures which will help us understand co-productive processes' (p7).

Osborne *et al*,[8] starting from the premise that co-production is (or has become over time) a 'woolly word', helpfully critiqued its conceptualisation within the policy and practice literature, linked it directly to the co-creation of value in public service delivery and proposed a framework for differentiating 'the cluster of related concepts that are contained within the term 'co-production'.[8] This framework comprises 'four ideal types of value that are co-created in public service delivery by the iterative interactions of service users and service professionals with public service delivery systems,' namely, co-production, co-design, co-construction and co-innovation. Voorberg *et al*[16] conducted a systematic review of the literature (1987–2013) relating to co-creation/co-production with citizens in public innovation across all sectors, concluding that most studies focus on the identification of influential factors, while hardly any attention was paid to outcomes. Greenhalgh *et al*[5] report a narrative review of different models of co-creation

relevant to community-based health services; they identify key success principles but note that 'impact is by no means guaranteed'.

For the purposes of this rapid evidence synthesis (RES), we identify co-production and co-design as specific approaches to co-creating value at the service (or clinical microsystem[13]) level within an acute healthcare organisation. Co-design is an approach to participatory design (although traditionally of a new product) that seeks to actively involve all stakeholders (eg, staff, patients, citizens) in a process to help ensure the result meets their needs and is usable. Such approaches are increasingly being used, often situated in the emerging service design field[17 18] as part of co-production projects. Specific forms of co-design such as experience-based design (EBD)[19 20] and experience-based co-design (EBCD)[21] have been developed and applied in the healthcare sector. We also follow Osborne *et al*[8] conceptualisation of their final two 'types of value' (co-construction and co-innovation) as being concerned with service systems (rather than individual services)[8]; we return to this important distinction at the end of this paper.

We undertook this RES partly to inform an ongoing study (Collaborative Rehabilitation Environments in Acute Stroke (CREATE)) which is bringing together staff, former patients and carers to review and jointly co-produce the way in which therapeutic rehabilitation-related activity is provided in stroke units in the early days and weeks after stroke.[22] In the absence of existing published evidence, this RES aims to identify and evaluate reported outcomes of co-produced interventions designed to achieve patient-focused quality improvements in acute healthcare settings. To inform the empirical study, our secondary objectives were to identify in such settings:

► those elements of co-production processes that lead to patient-focused quality improvements;
► any evidence of cost-benefit analysis, cost reduction or other economic evaluation associated with co-produced interventions;
► barriers and facilitators reported to influence the implementation and use of co-production methods and implementation of co-produced interventions;
► issues which impact sustainability of quality improvements following the use of co-production.

## REVIEW METHODS

While there are no agreed international guidelines for the design and conduct of RES,[23] there is overall agreement that the process involves providing an overview of existing research on a defined topic area, together with a synthesis of the evidence provided by these studies to address specific review questions. RES are typically completed in 2–6 months; this does not normally allow for all stages of traditional effectiveness reviews.[23–25] However, RES are increasingly valued by policy makers who require rapid answers to specific questions to inform policy revision or service development.[26 27]

---

**Box 1  Database and hand searches**

Medline, Cinahl, Web of Science, Embase, HMIC, Cochrane Database of Systematic Reviews, SCIE, Proquest Dissertation and Theses, EThOS and OpenGrey.
The following journals were hand searched: *CoDesign*, *The Design Journal*, and *Design Issues*.

---

**Box 2  Inclusion criteria**

► Reports research using a co-creation or co-production or co-design or experience-based co-design approach in an acute healthcare setting.
► Reports patient or staff or organisational outcomes resulting from research using a co-creation, co-production, co-design, experience-based co-design approach in an acute healthcare setting.
  Outcomes of interest include the following:
  ► any measure of the outcome of co-produced interventions on patient-focused quality improvements in acute healthcare settings as reported by patients or families or caregivers or health service providers;
  ► including patient-reported outcome measures and patient- or staff-reported experience measures;
  ► using qualitative or quantitative data.
► Acute healthcare settings include the following:
  ► emergency departments/accident and emergency departments;
  ► adult inpatient facilities including: acute medical or surgical admission units (often termed MAUs or SAUs), acute medical or surgical units, acute trauma units, acute neurological units, intensive or critical care units, acute care of the elderly or geriatric units, medical oncology or cancer services;
  ► adult outpatient facilities including medical, surgical, trauma, neurology, care of the elderly or geriatrics, medical oncology or cancer services.

---

Limitations of RES include the potential for introduction of bias. Sources of bias can include omission of critical appraisal of methodological quality of included papers. Similarly, limiting the scope of the search to published literature or specific databases, to a specific focus within a larger topic or to a reduced timespan of publication can reduce the applicability of RES findings for some audiences. In RES of effectiveness, meta-analyses may be omitted. It is important therefore that RES are conducted systematically; methods used and limitations introduced by these should be reported transparently.[23–26]

This RES was conducted between January and June 2016. We used a systematic approach and included full-text evidence published in English within the last 10 years. Search terms and databases searched were agreed by researchers and information specialists at the University of Leeds. Search terms used were specific to the use of co-production in acute healthcare settings (see box 1 and online supplementary file 1). To keep the search focused, we omitted broader search terms including cooperative behaviour, patient participation, collaborative approach and service improvement.

Database searches were conducted for the period 1 January 2005 to 31 January 2016. Given existing reviews[5 16] and the CREATE study focus, we reviewed post 2005 evidence and only that reporting on studies in acute healthcare settings. We completed citation tracking of five seminal papers; five experts in co-production were requested to nominate three to five seminal papers relevant to our review. Inclusion criteria are shown in box 2.

Two reviewers independently read all titles and abstracts. Discrepancies in retain or reject decisions were addressed by discussion between reviewers with recourse to a third where consensus could not be reached. Three reviewers independently read the retrieved full-text papers; initial decisions to retain or reject were made independently based on the inclusion criteria. All three reviewers then discussed retain and reject recommendations and reached a consensus agreement.

Three reviewers completed data extraction. We used quality appraisal checklists developed for quantitative and qualitative studies[28] (box 3). These were selected as they were developed and used by the National Institute for Health and Care Excellence (NICE) in the appraisal of the quality of evidence underpinning NICE public health guidance. The checklists address fourteen areas of study quality ranging from theoretical approach through study design, data collection and analysis methods and also ethical review (see online supplementary file 3, quality

assessment records). For each study, two reviewers undertook data extraction and quality appraisal independently. Studies were not excluded on the basis of quality appraisal; this was used to inform the discussion of the evidence identified in the review. We adopted a mixed research synthesis approach[29] drawing on methods reported by Boger *et al.*[30] We used an integrated design which entailed grouping studies for synthesis not by methods (ie, qualitative and quantitative) but rather by findings viewed as answering the same research questions or addressing the same aspects of a target phenomenon.

## RESULTS

A total of 712 titles and abstracts were identified for screening from which 24 papers were identified for full-text review. Following review, 11 publications were included in the final evidence synthesis (figure 1); full details of each study can be found in the online supplementary file 2.

Papers were published between 2008 and January 2016, with studies conducted in five countries (Canada,

---

**Box 3  Quality assessment study rating criteria[28]**

► ++ All or most of the checklist criteria have been fulfilled; where they have not been fulfilled, the conclusions are very unlikely to alter.
► + Some of the checklist criteria have been fulfilled; where they have not been fulfilled or not adequately described, the conclusions are unlikely to alter.
► − Few or no checklist criteria have been fulfilled, and the conclusions are likely or very likely to alter.

---

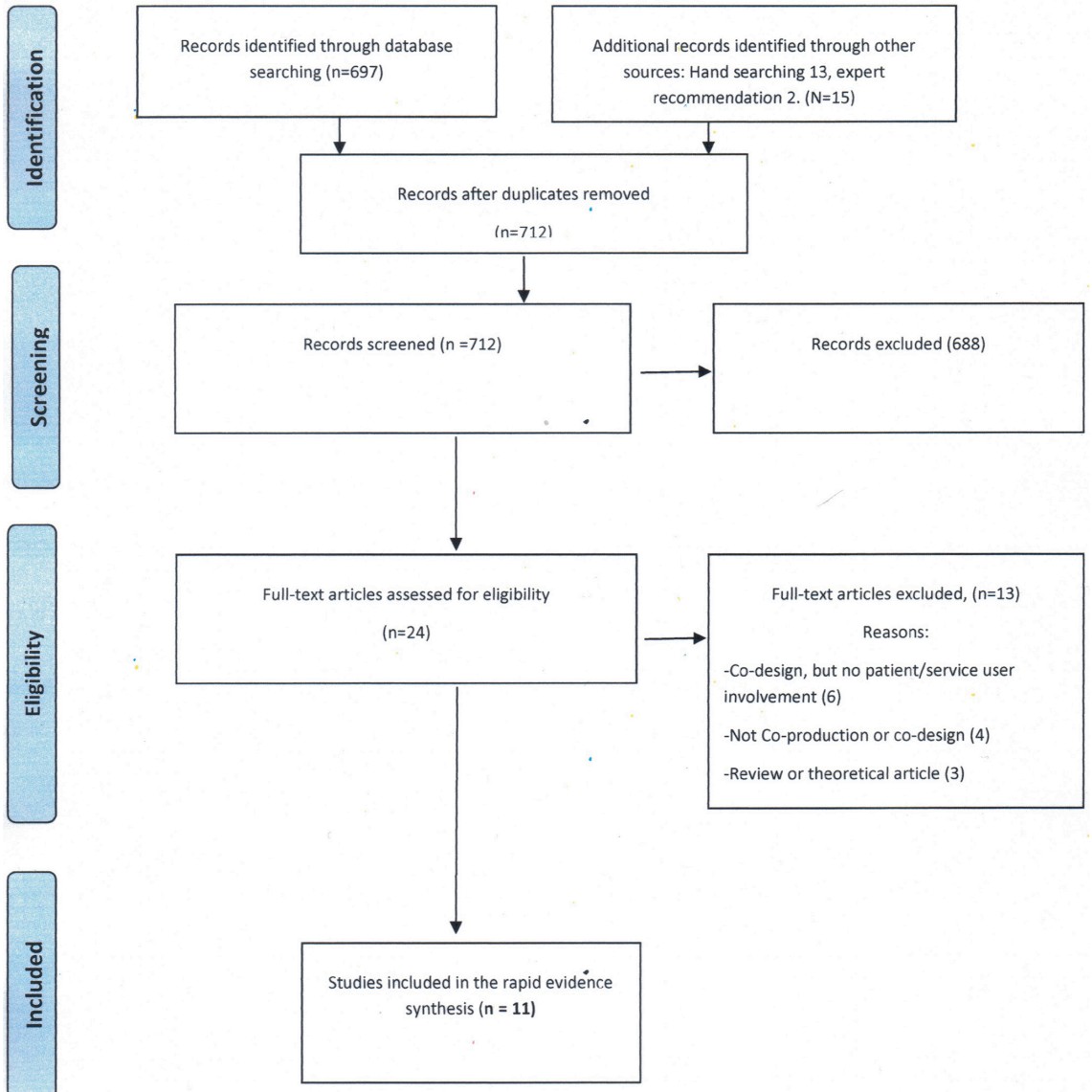

**Figure 1** Adapted Preferred Reporting Items for Systematic Reviews and Meta-Analyses (PRISMA) flow diagram.

England, New Zealand, Australia and the Netherlands). Settings included intensive care units (n=1); inpatient and outpatient oncology services, including breast, lung, colorectal and haematology services (n=5); mental health services (n=1); emergency departments (n=1); an outpatient clinic for people with multiple sclerosis (n=1); and older peoples outpatient services (n=1). A further study evaluated co-design projects in five Dutch hospitals; the projects were conducted in four oncology departments and a haematology department.

One study was a feasibility randomised controlled trial (RCT) of a co-designed intervention.[31] The study design was explicitly stated in only one of the remaining papers[32]; the remaining studies used descriptive qualitative approaches to evaluate changes in services or explore participant's experiences or views, although these were rarely described in detail. Data collection methods included self-report postal questionnaires, ethnographic observations of patient journeys through services and of

staff working practices, semi-structured interviews with staff and patients (sometimes filmed), focus groups and emotional mapping exercises. Data analysis techniques were described in any detail in only 6 of the 11 publications.[32–38] Most papers focused on the processes used to understand and co-design within services rather than evaluating outcomes; the exception being the feasibility RCT.[31]

Quality assessment ratings are shown in the final column of the table in the online supplementary file 2. Four studies were rated at ++ with the remaining seven at +. Ratings largely reflect the omission of detail on research methods, particularly in relation to stating research aims, questions, sampling decisions and discussion of data analysis and findings.

### Use of co-production methods

Methods reported included a feasibility RCT of the impact of a co-designed intervention,[31] participatory

design,[38] creative design,[39] co-production,[37] EBCD,[34–37 40] accelerated experienced-based co-design (AEBCD)[32 41] and EBD.[33 42] EBD or variants of this approach were reported in 8 of 11 included studies, with a further study[31] testing an intervention previously designed using EBCD.[40] Projects ran for 3 months to over 1 year. While local tailoring of approaches was clear, the use of a staged process for co-production was evident across all studies. This indicated that while the classification of approaches varied, the majority demonstrated patient, carer and staff reflecting and working together using participative methods in joint review and co-design activities focused on improving patient and/or carer experiences.

There was variation in levels of participant engagement. In an oncology service, while patients and a wide range of staff were included in the creative design project, they participated separately in the three stages of the study and did not come together to share and discuss their respective experiences.[39] In the Thomson *et al* study, a designer-led, three-stage process was used in which carers and staff came together only once; this variation in participant engagement did not appear to have any impact of the range of practical service improvement outcomes.[38] However, in the case of the Golden *et al* study,[39] the lack of joint working among the patients, carers and staff may have been a key reason why no specific outcomes or agreed plans to introduce the changes identified in the creative design process were reported.

### Reported outcomes of co-produced interventions

Our primary aim was to identify and evaluate the reported outcomes of co-produced interventions designed to achieve patient-focused quality improvements in acute healthcare settings. The methods used to understand participants' experiences and obtain their views included observation, process mapping, interviews, focus groups and postal survey; more detail is reported in the online supplementary file 2. Overall, patients reported positive experiences of participation, consistent with those associated with a wide range of co-production in the public and healthcare sectors.[5 7 10 12 16] Co-production or co-design projects were effective in generating a wide range of ideas and specific suggestions related to improving patients' experiences across the different settings. Reported service changes and quality improvements are summarised in table 1. However, clarity regarding implementation of improvements to services and/or evidence of resulting impact on patient outcomes was one of the main limitations of the reporting in the majority of the studies.

The feasibility RCT[31] evaluating the impact of a co-designed intervention to improve the knowledge, experience and emotional well-being of carers of people with breast, lung or colorectal cancer was the only study reporting use of a validated outcome measure, the General Health Questionaire (GHQ)-12.[31] The other quantitative measures used in this study were adapted from existing measures (n=2, eg, supportive care needs) or developed by the study team (n=2, eg, experience of care and knowledge

of chemotherapy). Carers who received the intervention reported significantly better understanding of symptoms and side effects and that their information needs were more often met compared with the control group.[31] Confidence in coping improved between baseline and follow-up for the intervention group and declined for the control (although differences were not statistically significant). Accepting the limitations of the single centre and small sample size, this study provided evidence for the effectiveness of the carer-focused intervention developed in an earlier study by carers and staff using EBCD methods.[40] In the remaining 10 studies, there were no reported instances of use of validated outcome measures, such as quality of life measures, nor any instances of validated patient-reported experience or outcome measures.

Four studies[33 35 37 39] did not clearly differentiate between the findings from the co-production or co-design processes used and patient-focused quality improvement outcomes. In these studies, it was difficult to distinguish between ideas for improvement generated by co-design activities and tangible service improvements that had either been introduced or were planned. However, the reported 'outcomes' of the studies can be categorised in three main ways and are summarised in table 1:

1. patient and staff involvement in the co-production or co-design processes;
2. generating ideas and suggestions for changes to processes, practices and clinical environments impacting on patients' and/or carers' experiences of a service, (and often indirectly on staffs');
3. tangible change in services and impact on patient or carer experiences.

### Elements of co-production that led to quality improvement

Not all authors commented directly on the process of co-production. Those that did identified that co-production methods provided the means to ensure patients' experiences, concerns and ideas for change were captured, presented to and discussed with staff in services. Active engagement of patients in this way gave both legitimacy and urgency to service improvement plans,[36 37] particularly where managers were involved in or actively supported co-production projects.[35 37 41 42] Tsianakas *et al*[36] identified genuine and direct patient and carer involvement (relative to other service improvement projects in which patients and staff had participated) and linked this to patient participants taking direct responsibility for the work and its outcomes as being success factors linked to the EBCD process itself. Similar claims were evident in other EBD and EBCD projects,[32 33 37 42] where patients perceived there to be a partnership with staff in the co-production process[32 33] and where staff reported facilitated co-design groups directly contributed to active and empowered partnerships between staff and patients in which provided genuine opportunities for patients to 'have their say'.[32 33 37] There appeared to be an equally powerful impact on some staff participating

**Table 1** Reported outcomes of co-production in acute healthcare settings

| Type of outcomes | Evidence synthesis | Studies4 | Example |
|---|---|---|---|
| Patient and staff involvement | Patients perceived being really listened to by staff, having the sense that involvement was genuinely collaborative and that staff were committed to working with patients to improve services. | 32–34 36 41 | Key to success was thought to be the strong relationship between patients and staff that has been built over time.[36] 'Co-production' approach was seen to give patients, carers and staff a deeper understanding of each others' experience of the processes of care. It provided an opportunity to work meaningfully together, and this strengthened relationships among those involved in EBCD.[34] "It was good, it felt as though [the hospital] and the professional staff were really interested … There was a comradeship if that's the right word … " (older patient—33p) |
| | Patients and staff recognised levels of participation and collaboration were distinct from any quality improvement activity they had experienced previously. | 33 34 36 37 | EBD's techniques facilitated building trust and rapport[33] EBCD was perceived as a significantly higher level of genuine direct patient and carer involvement (relative to other service improvement projects in which patients and staff had participated)— "the constant feedback [being] really, really useful", and "[needing] the eyeballing of each other to make it work"[36] The EBCD process was thought to have given patients a greater sense of direct responsibility for the work and its outcomes[36 37] and led to an increased understanding of the 'other' perspective, resulting in a potentially broader cultural change in mind sets and behaviour. Staff participating in the co-design groups reported a greater sense of empowerment to make changes to their service.[36] Allowed nurses to reconnect with their core values of caring and compassion; development of a sense of empathy with patients was also evident. |
| | Staff realised that they often really did not have a good understanding and insight into patients' experiences of the services they provided. | 36 37 41 42 | Letting their 'own' patients talk about their experiences in a common 'action' setting affects professional staff, as patient stories make visible what the impact of (already known) problems really is on patients. The active role of patients and the way they express their experiences create a sense of urgency to act on the improvement issues raised.[37] Staff identified that working with patients in this honest and open sharing process was moving and at times humbling.[36] |
| | There were also negative perceptions reported by a small number of patients and staff. These related to questioning the value of the amount of staff and patient time invested and to not being able to see or experience actual service changes. | 33 34 | "'[The project] Seemed to go on for a long time …' and I don't think we were properly prepared for that. … my initial impression was that it would be perhaps a couple of interviews, and a discussion" (older patient—33p). Both patients and staff identified that some service changes did not happen, for example, "not sure if it has had any effect" (patient), "in some areas, the area has not been improved" (staff, "there's a few changes. … [but] … there was some really serious things … they were highlighted, there was no resolution" (staff).[33] Authors noted that although consensus was reached quite easily, staff groups had tendency to focus on more positive aspects of care and recent developments/improvements, whereas service users and carers were more openly dissatisfied.[34] |
| | Negative staff experiences and perceptions were also evident; these all related to challenges in conducting co-production or co-design projects. | 33–35 37 42 | Staff (and patients) reported little or no change had occurred. "there's got to be something at the end of it" (staff member—33p). Staff viewed EBCD as a burden, additional to an already busy schedule.[34 37] Staff workloads increased where EBCD was additional to usual roles.[33 35 37 42] Lack of time and high-level organisational support for steering group to complete implementation plans.[35] Lack of organisational capacity to respond to 'bottom up' service development.[35] Fee-for-service systems do not directly reward (or encourage) EBCD/co-production projects.[37] |

**Table 1** Continued

| Type of outcomes | Evidence synthesis | Studies4 | Example |
|---|---|---|---|
| Generating ideas and suggestions for changes to processes, practices and clinical environments | The number of ideas generated or issues highlighted ranged from 5 to 11 core issues, through 48 co-design activities to 400 ideas generated by staff in an oncology service project. | 33 35–37 40 42 | New template for patient letters.[33] A patient journey map and accompanying journey guide were developed linked to a patient-held record.[42] |
| | A significant number of factors affecting patient experience from across the range of acute healthcare settings related to the structure and process of care and service provision. Changes spanned: -information and communication processes -environmental and organisational changes (and to a lesser extent) -staff training and development. | | A clinic guide and clinic (terms) dictionary were developed for new patients.[38] Redesigned discharge summary, improved cross site information booklet for patients transferring to another hospital for surgery.[32] Patient-focused visual journey guide.[42] Proposals for layout of roads surrounding the outpatient building, design proposals for new way-finding materials (signage and maps).[33] Guidance on diagnosis procedures included in junior doctors' induction.[36] Informal staff training programme—'recent developments in'.[38] A co-design toolkit and website for other services in New Zealand.[42] |
| | Acute hospital admission and care have impacts beyond individual patients; involvement of families and carers—evident in five of the studies—appears to have helped ensure that those involved in the patients' journey were also able to comment on and shape that experience. | 31 35 36 | Administrative staff receive customer care training watch the patient experiences (DVD),[33] Ensuring that when carers came to visit, they were greeted appropriately at the door of the unit.[32] Clinic structures changed to reduce waiting times.[33] Carers' information leaflet increases knowledge and understanding of symptoms and side effects and carers' confidence.[28] |
| | The structured co-production and co-design methods employed empowered both patients and staff to collaboratively review services and to plan and undertake changes in their workplaces. | 32–34 36 38 42 | Despite concerns about activities being 'a bit daunting at first', participants supported researchers' perceptions that sharing experiences via storytelling and emotional mapping helped to build empathy and cohesion in the project group.[30] The opportunity to hear directly from patients and carers reportedly had a transforming effect on some staff. "Our data suggest face-to-face encounters with patients in co-design groups often had a profound effect on staffing making them think differently about their practice and reconnect with their core professional values, resulting in renewed motivation".[29] |
| | Involving designers/design researchers in co-production projects added value, although challenging staff and patients way of thinking about their healthcare journey and experiences and how these could be improved within existing service settings. However, as design-led techniques may be unfamiliar to staff and patients, these and their purpose need to be carefully explained if they are not be seen as time wasting or games. | 33 38 39 | Forum theatre training to improve staff awareness of 'customer care'.[33] Use of metaphor, analogy and prototyping techniques (travel documents linked to an ideal patient journey). The props and the analogy facilitated the co-design process by enabling participants to not just suggest solutions to problems but create and share more radical suggestions for change (such as initiating a patient advisory group).[38] Photojournaling and storyboarding[39] |
| Tangible change in services and impact on patient or carer experiences | Patient and carer experience: overall, patients reported positive experiences of participation. Despite the barriers identified, participants across settings viewed the benefits of patient and staff involvement in structured co-production/co-design projects as outweighing the challenges. | 31 32 34 36 42 | Carers who received the co-designed intervention reported significantly better understanding of symptoms and side effects and that their information needs were more often met than the control group. Confidence in coping improved between baseline and follow-up for the intervention group and declined for the control.[31] Patients can opt to receive chemotherapy treatment information in a group with others having the same treatment.[36] More comfortable V-shaped pillows for postoperative patients.[32] Changed process for porters to remove waste avoiding ICU rest times.[34] Provision of free refreshments (patients and families).[42] New design for mammography gown.[42] |
| | Service changes (processes and environmental change): it was not always possible to distinguish changes that had been implemented from those which were planned. Reporting of service improvement outcomes per se was a limitation of the included studies, but the projects reported largely indicated that the structured co-production and co-design methods employed empowered both patients and staff to collaboratively review services and to plan and undertake changes in their workplaces. | 32 34 36–38 42 | Introduction of mini-Schwartz rounds on ICU.[32] New private room identified for receiving support after diagnosis.[32] Installation of a new air conditioning unit in the waiting room, installation of a blanket warming cupboard, installation of payphones in waiting areas.[34] Changes to structure of clinics to reduce waiting times.[36] Decoration of waiting rooms.[37] Volunteer in the outpatient clinic linked to a patient advisory group.[38] Psychosocial care offered proactively during consultations (one hospital).[37] Patient held record to track appointments.[42] |

DVD, digital video disc; EBD, experience-based design; EBCD, experience-based co-design; ICU, intensive care unit.

in co-production in that the methods required them to engage with and listen to patients in ways most had not experienced previously.[34 36 37 42]

However, despite clear evidence of the contribution of co-design activities in generating ideas for patient-focused service improvements (see table 1), in some projects, staff and patients were frustrated by the lack of progress from problem and solution identification to actual quality improvements.[33–35 37] In one case, the funded project appeared to end before concrete action planning to effect change, based on the ideas generated, had occurred.[39] Researchers ascribed this to a combination of the time taken to undertake co-design activities and the often slow processes of change in complex health service organisations.

### Economic evaluation
There were no cost–benefit analyses related to reported outcomes for any of the studies. Only one study reported a cost analysis of co-production methods by comparing the cost of an AEBCD approach to standard EBCD.[32] A significant reduction in costs was reported for AEBCD. The cost of developing patient experience (trigger) films fell from £30 485 to £8289, and as the film developed could be used in future projects, this was a one-off cost. This study was also the only one to quantify costs of the co-design stages which were £20 276; over half of this sum was for the salary of local facilitators.

Other authors alluded to the costs associated with participatory or co-design methods in terms of the largely time-related costs involved in freeing staff to take part in or run the projects or costs associated with actual or planned service improvements. These include, for example, improved signage,[33] provision of personal care equipment,[34] information booklets/leaflets, digital video discs,[32 36–38 42] a website and web-based toolkit[42] or costs associated with having blood tests conducted off site.[39] However, no actual or estimated costs were reported for these elements. Similarly, 9 of the 11 publications reported on researcher-led, designer-led or researcher-supported projects but in only two cases were actual research costs traceable.[32 39]

### Barriers and facilitators to implementation
The 11 studies encompassed nine projects conducted in diverse acute healthcare settings, although few of the reported barriers or facilitators were specific to project or setting. The most commonly reported barriers encountered in using co-production approaches in acute healthcare settings included a lack of support, resources or managerial authority to bring about structural or environmental changes.[33–35 37 42] In addition, practical or logistical problems were identified, for example, ensuring frail elderly people could attend regular co-design meetings.[33] Recruiting patients and carers and retaining them through the different stages of projects were highlighted by a number of researchers.[32 34 35 37 41] In at least one project,[34] some patients participated only

in discovery or exploratory phases, then having 'shared' their experiences with staff, declined further participation arguing that they were no longer using the service. Researchers also identified the need to plan for and manage patients' understanding of what may be a radically different form of engagement with hospital staff, often quite unlike that experienced previously by users of health services.[32 33 36] In the studies reviewed, the frequency and duration of their involvement and also the time it may take to bring about changes in the structures, processes and sometimes the physical environments of services were highlighted as factors to be addressed with participants.[33 39 42]

In the majority of projects, staff engagement was in addition to usual clinical or managerial roles; nonetheless, a high level of interest in and commitment to co-production activity was identified in almost all projects. However, this was impacted on in at least five projects by staffs' frustration at the expectation that they would undertake co-production/co-design work in their own time, that they could not allocate time out of their routine work and that additional support was often not provided by more senior staff.[33–35 37 42] For some staff, this made participation almost impossible[37]; for others, it meant projects did not progress as expected[35] or contributed to tensions in co-design groups or between researchers and participants.[33] The duration of projects also increased the likelihood that staff turnover would impact on project leadership or involvement.[34 35] The need for structured and ongoing managerial and organisational support was highlighted,[32–34 36 38 41 42] but only two studies expressly refer to governance or oversight groups set up to support co-design projects.[33 34]

In projects where facilitators were engaged formally and funded to manage or oversee projects, it was more likely that projects (A) maintained momentum and were delivered as planned, (B) engaged and retained patients, carers and staff and (C) generated concrete examples of areas where patients' or carers' experiences could be improved.[32 35 36 41] In some studies, researchers (some of whom were designers) facilitated staff and patient engagement in the projects. Where designers were directly involved, our review suggests that they introduced ways of thinking and working which successfully challenged staff and patients to reconceptualise everyday processes and activities. This was achieved using metaphor games, design experiments, visual storyboards, prototyping, future focus groups and emotional mapping, approaches not familiar to most health service staff.[32 33 38 39] However, with the exception of Locock *et al*,[32] the costs associated with facilitator or researcher involvement were not identified.

Despite the barriers identified above, the studies suggest researchers and participants across settings viewed the benefits of this level of patient and staff involvement in structured co-production/co-design projects as outweighing the challenges.

## Sustainability

None of the 11 studies formally evaluated whether co-production or co-design as a way of working had been sustained or whether improvements made as a result of such approaches were sustained over any length of time. However, a number of concrete changes in service organisation, care environments and patient and in carer experiences occurring either during or closely related to co-production activities were reported (see table 1).[33–38 42] Bowen et al[33] indicate that some changes may take 2 years or more to effect (eg, changes in hospital signage and routing), a period of time long after staff and patients had ceased to be actively involved in specific projects; another study reported that planned changes may not be achieved at all despite extending the life of projects.[35]

## DISCUSSION

This rapid evidence synthesis has summarised peer-reviewed evidence relating to the use of co-production approaches in focused quality improvement projects in the acute healthcare sector.

Different forms of co-production/co-design were most commonly reported, and few criticisms of these methods were identified, in part reflecting an almost certain positive publication bias. However, critical reflections are emerging.[37 43 44] These, in part, explore the so-called 'dark side' of co-production recognising, for example, the potential for reproducing inequality as those who are most able to are those that co-produce; the risks of 'captured' co-production where participants are compelled, sometimes unknowingly, to co-produce; co-production as a substitution of labour (what Fotaki terms 'a race to the bottom in times of austerity'[43]); and co-production being appropriated as cover for inherently political decisions.

The kind of patient-focused quality improvements reported in the identified studies can largely be considered examples of simple practical changes or 'sweating the small things'.[45] However, from the perspective of patients' experiences of and journeys through a service, these issues were likely to make care and treatment more humane and more person centred, as well as making journeys easier to engage with and negotiate. However, both Bowen et al[33] and Thomson et al[38] argue that the ideation tools in EBD and EBCD were limited and that incorporating 'designerly' thinking into co-production projects could stimulate greater creativity and alternative solutions in sometimes rigid health services.

The relative lack of robust evaluations we found may also reflect the stage of adoption of these types of methods in acute healthcare settings as we report there is currently only one published feasibility RCT. However, there is a large-scale stepped wedge RCT under way in nine Australian community mental health teams which is evaluating the effect of co-design on psychosocial recovery outcomes (such as willingness to ask for help and personal confidence and hope).[46] In future studies, robust and routine evaluation of the costs and impacts of using co-production methods are required so that those leading and participating in quality improvement efforts in acute healthcare organisations can make informed decisions regarding their adoption in addition to (or instead of) other quality improvement methods. However, reliance on relatively broad and insensitive measures—such as patient satisfaction surveys—may overlook the real value placed by patients and staff on changes in the personal behaviour, attitudes and culture of healthcare teams. The studies reviewed here consistently referred to these service level benefits of co-production for participants and services, as well as pointing to potential service system level effects such as those identified by Osborne et al.[8]

As others have highlighted, the use of similar approaches in the community healthcare sector has not always 'follow(ed) through to significant and sustainable redesign'.[5] A recent narrative review of four different models of co-creation relevant to community-based health services proposed three 'key success principles', namely, (1) a system's perspective (assuming emergence, local adaptation, and non-linearity), (2) the framing of research as a creative enterprise with human experience at its core and (3) an emphasis on process (the framing of the programme, the nature of relationships, and governance and facilitation arrangements, especially the style of leadership and how conflict is managed). The review authors proposed that co-creation 'failures' could often be tracked back to abandoning (or never adopting) these principles.[5] The publications in our RES do not specifically identify these principles and provide no clear evidence that such principles underpinned the planned research and service improvements noted. However, individually and collectively, included studies highlighted problems which arose when explicit adoption of such principles was not evident in the set up and delivery of the reported co-production projects in acute healthcare settings. Although the acute healthcare sector faces challenges associated with rapid patient turnover and with patients sometimes critically or terminally ill, the majority of studies in this RES demonstrate that co-production is possible and was valued by participants. These findings have helped in planning for the introduction of the EBCD approach in the CREATE study. Although none of the studies reviewed were conducted in stroke services, the facilitators and barriers identified in the RES, together with the principles identified in the Greenhalgh et al review,[5] indicate key areas for research or service improvement teams to address with managers, staff, patients and relatives participating in the EBCD project activity. Specifically, future studies should evaluate clinically related outcomes, for example, through validated patient-reported experience and outcome measures. Cost-effectiveness of co-production projects can and should be evaluated using standard health economic approaches. Equally important is the impact of

service change on factors such as patient access to and flow through services and evaluating whether changes are sustained overtime. Potentially broader impacts on the values and behaviours of participants, including staff, are more difficult to measure but could be evaluated qualitatively and quantitatively and incorporated in the evaluation of co-production projects in acute healthcare settings.

Co-production is an emerging focus for research and evaluation in the health sector. Currently, there is no international agreement on or consistent use of terminology to capture the range and diversity of participative approaches increasingly employed in health services worldwide. However, we acknowledge that the exclusion of broader search terms relating to patient participation, patient centredness, service improvement and clinical microsystems is a limitation of this rapid evidence synthesis. We limited search terms to those we anticipated were likely to identify research studies reporting on use of recognisable and structured co-production or co-design approaches. In so doing, we did not access the more diverse and extensive literature reporting on projects including patient or service user participation or projects which may be termed 'patient or user centred'. This literature commonly includes descriptive summaries, reports and commentaries; while these descriptions can be rich and informative in terms of service review and development, the publications often do not report on research or structured service improvement methods. We did search for evidence in a database of grey literature from Europe (Open-Grey) but identified no publications meeting our inclusion criteria. Our strategy of asking five experts in the field to nominate key publications was designed to reduce the likelihood we did not identify key research publications. Nonetheless, it is possible that the narrow focus of the search terms used in our rapid evidence synthesis means we did not identify research and evaluation of co-production projects in the health sector reported using some of the alternative terms described above. Such literature may have been consistent with or challenged our findings. However, this evidence synthesis drew on good practice guidelines for such reviews and was rigorously conducted.[23][25][26] This study is the first to systematically review outcomes associated with developing and implementing co-produced interventions in acute healthcare settings.

While the potential benefits of using co-production in acute healthcare settings have been identified, these must be set against the concerns identified by participants in the studies we reviewed that such approaches can be challenging to set up and implement in busy clinical environments where no formal, practical and financial provision is made for staff, patient and carer involvement on a sustained basis. Accelerated approaches may offer a partial solution but organisation-level understanding of and commitment to these kinds of service improvement methods—as also

identified by Greenhalgh *et al*[5]—remains key to their likely effectiveness.

## CONCLUSIONS

We have identified a lack of rigorous evaluation of effectiveness and cost-effectiveness of co-produced interventions in the acute healthcare sector at both the service and system levels. Traditionally, co-production has been applied in community settings, but there is growing recognition of the value of a more collective contribution to quality improvement work from those who are both delivering and receiving any form of public service.[12][47] At the clinical microsystem level within the healthcare sector, there has been increasing adoption of co-production involving patients and staff, typically drawing on co-design approaches, as a means of improving the quality of front-line services.[12][13] Health services, whether public or private, operate within increasingly severe financial restraints where funding for quality improvement work is limited. Without robust critique, evaluation and evidence of the co-creation of value, there is a danger of co-production becoming another management fad or fashion,[48] with the meaning of the term itself appropriated and co-opted but no longer 'co-produced'.

**Acknowledgements** The arguments presented in this paper were partly informed by discussions during a workshop at the University of Nottingham on 7–8 July 2016 organised by Professor Andrew Gray and Professor Bruce Stafford: 'The Individual and the State: co-production and interdependency in the mixed economy of public service consumption'. Our thanks go to Deirdre Andre and Sally Dalton, information specialists at the University of Leeds who helped refine the search terms and strategy and conducted the database searches. Tino Kulnik assisted with the initial screening of abstracts. Stephanie Honey and Parminder Dhiman, research fellows, also assisted with some of the data extraction and quality assessments.

**Contributors** DC, FJ, RH and GR designed the rapid evidence synthesis and developed the search terms. GR conducted hand searches of the journals: *CoDesign*, *The Design Journal* and *Design Issues*. DC and FJ screened all abstracts and agreed on full text for review. DC, RH and FJ reviewed all full texts and agreed inclusions and exclusions. DC, FJ and RH independently completed data extraction and quality assessment for all included papers. DC prepared the manuscript, and GH, RH and FJ reviewed and revised the manuscript. All authors agreed the final version of the manuscript.

**Funding** This project is funded by the National Institute for Health Research (NIHR) Health Services and Delivery Research Programme (project number 13/114/95). The views and opinions expressed therein are those of the authors and do not necessarily reflect those of the Health Services and Delivery Research Programme, NIHR, National Health Service or the Department of Health.

**Competing interests** None declared.

**Provenance and peer review** Not commissioned; externally peer reviewed.

**Data sharing statement** No additional data are available.

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
