## [Reviewer comments · BMJ Open]

ARTICLE DETAILS

TITLE (PROVISIONAL)	What outcomes are associated with developing and implementing co-produced interventions in acute healthcare settings? A rapid evidence synthesis.
AUTHORS	Clarke, David; Jones, Fiona; Harris, Ruth; Robert, Glenn

VERSION 1 - REVIEW

REVIEWER	Dr Gemma Stacey University of Nottingham, UK
REVIEW RETURNED	17-Oct-2016

GENERAL COMMENTS	Many thanks for the opportunity to review this excellent example of RES in the complex field of co-production. The rationale for both topic and method was comprehensive and transparent. The findings are reported in line with the project aims and discussed in relation to the wider evidence base. Limitations are acknowledged and conclusions are well founded. In terms of development I would like to suggest some description of the co-produced interventions included in the papers. You identify significant variance in the types however the practical enactment is not included. Also some consideration into how this variance may influence outcomes could be discussed ie what is considered a co-produced intervention is far from standardized.
---

REVIEWER	Daniel Wolstenholme NIHR CLAHRC YH, England
REVIEW RETURNED	31-Oct-2016

GENERAL COMMENTS	Thanks very much for this timely and thoughtful paper. It resonates with our frustrating experience of trying to find published work across the design and health literature describing co-design/ co-production. I do think that there might be some work that uses 'co-production' but doesn't recognise or describe it in such a way (the Clinical Microsystems work from Dartmouth) which would not have appeared in your search, but would have had more of a focus on measurement of patient or service outcomes. I do think there is something about the co-production approach and it's attention to context and locally derived solutions that makes smaller projects harder to evaluate (under a traditional hierarchy of evidence), but that we should welcome the evaluation of methods/programmes with a sophisticated attention to a range of outcomes, perhaps with a realist focus.
--

	We should use this paper as a call to arms to support teams to evaluate and describe their projects in such way as to the true cost and benefits of the methods to be recognised to support the continued use (or not) of these methods in health and social care. On a very minor note the US spellings of recognize and other 'izes' are used throughout the document.
--	--

REVIEWER	Alison Eastwood CRD, University of York, UK
REVIEW RETURNED	05-Dec-2016

GENERAL COMMENTS	This manuscript describes a rapid evidence synthesis to identify outcomes associated with developing and implementing co-produced interventions in acute health care settings. The manuscript is well written and clearly describes the existing evidence identified and also provides recommendations for future research. The limitations of the work and implications of this for the findings presented are also described. I have a few minor comments which could improve the clarity of the manuscript:  1. p6, lines 20-21. Clarify how decisions on full text papers were made. Should retained full text papers" be "retrieved full text papers"? Did all three reviews independently make decisions on all retrieved papers? How were final decisions on inclusions arrived at? 2. p6, lines 23-24. A bit more information on quality assessment would be helpful. It is not clear from the text what is included in the checklists and why they were selected (the relevant checklists could also be added as supplementary material). 3. p6, line 56. Should (n=4) be (n=1)? 4. p7, Reported outcomes of co-produced interventions. More information on any difference in how outcomes were measured would be helpful (e.g. methods for obtaining patients and carers views) 5. p8, line 44. Typo - missing "to" (...required them to engage...) 6. p8, line 49. Typo - missing "]" on reference 34. 7. p22, Prisma diagram. The number of records excluded at Stage 1 doesn't add up, given the number of records screened and the number of full text articles assessed.
--

REVIEWER	Kim Peterson VA Evidence-based Synthesis Program Coordinating Center VA Portland Health Care System USA
REVIEW RETURNED	09-Jan-2017

GENERAL COMMENTS	Thank you for the opportunity to review this rapid evidence synthesis (RES) on developing and implementing co-produced interventions in
---

acute healthcare settings. As I understand it, the primary objective of this RES was to evaluate outcomes of co-produced interventions in acute healthcare settings to inform CREATE study development. Unfortunately, this RES found very limited rigorous evidence. This included only 1 RCT of fair-quality that found positive effects on career and patient outcomes. However, authors state that ultimately the findings helped plan the EBCD approach in the CREATE study by helping indicate key areas for their research team to address.

In general, this manuscript could benefit from shortening and major editing to improve writing clarity - primarily in the form of reducing jargon. For example, in the Conclusion paragraph on page 11, there is the phrase, "At the clinical microsystem level within the healthcare sector..." If this means 'in acute health care clinics', the readers would benefit from shortening the verbiage as such. The following are some additional specific suggestions for refining the manuscript:

- 1) Abstract - Findings: Suggest revising to more completely summarizing findings. For example, outcomes evaluated are listed, but the actual results of the RCT seem to have been completely omitted. Suggesting listing a few of the major barriers and facilitators. Also, figure indicated 712 articles were screened rather than 710.
- 2) Strengths and limitations box on page 3: Suggest elaborating on these in the Discussion
- 3) Background - last sentence on page 3 is a run-on and should be split into two sentences.
- 4) Background - 3rd paragraph on page 4: Strongly suggest reducing the jargon. I read this paragraph numerous times and still found it difficult to understand.
- 5) Last sentence in first paragraph that cites evidence that policy makers increasingly value RES': Suggest adding recent relevant publication of using RES in a health care system: Peterson, K., Floyd, N., Ferguson, L., Christensen, V., & Helfand, M. (2016). User survey finds rapid evidence reviews increased uptake of evidence by Veterans Health Administration leadership to inform fast-paced health-system decision-making. *Systematic Reviews*, 5(1), 132.
- 6) Page 5 - 2nd paragraph: Limiting the scope to a specific focus would not widely be considered a "bias"; rather, this would limit applicability to wider audiences. Suggest rephrasing and adding additional biases common to RES such as widely discussed in the RES literature; e.g., limited critical appraisal, etc.
- 7) 'Elements of co-production that led to quality improvement' paragraph on page 8: Particularly for this paragraph - but elsewhere as well - suggest stating bottom line at beginning of section. I re-read this section a few times and was not able to decipher the main findings. The 'Economic Evaluation' section on the following page is a good example of starting the section with a clearly stated bottom line up front.
- 8) 'Barriers and facilitators to implementation' on page 9: 2nd paragraph lists a bunch of barriers. Then, the subsequent sentences seem to be describing specific examples of barriers. But, it is difficult to link the examples to the preceding categories listed. What are

	'practical or logistical' or 'recruiting' barriers examples of from the preceding list (i.e., "lack of support, resources or managerial authority to bring about structure or environmental changes"). (9) Table 1 - inclusion criteria on page 5: Outcomes listed are very vague. Suggest providing some specific examples for context. (10) Discussion could also benefit from more detail on clinical and research implications. Authors state that their RES findings helped them in planning their CREATE study by indicating key areas for their research team to address. But, that is very vague. Suggest providing at least one specific example for context. What key practical messages should others interested in co-producing interventions take away from your review? (11) Future research: Authors suggest that future studies should focus on clinical and cost-effectiveness - but how? Suggest providing some more specific suggestions in the Discussion. For example, on page 7, authors emphasize the lack of outcome measurement using validated tools. In the Discussion, suggest revisiting this issue by proposing not just more research, but more robust research in terms of what key outcomes to measure and how they should be measured. What is the bar that a researcher to strive to reach?
--	--

REVIEWER	Isomi Miake-Lye VA Greater Los Angeles, USA
REVIEW RETURNED	09-Jan-2017

GENERAL COMMENTS	The authors present an interesting synthesis of co-produced interventions, which is likely a topic of current interest to many readers. There are a few points the authors should consider: Abstract: There are a few grammatical issues (e.g., line 35 says "focus on clinical" but may need another word to clarify, clinical impact? clinical outcomes?). Both in the abstract and full text, articles are categorized as a feasibility RCT, 3 process evaluations, and 7 qualitative methods papers. This isn't quite parallel, since qualitative methods are not a study design. It would be more appropriate to say "descriptive qualitative" if these are not evaluations and only provide descriptive information (or describe the appropriate design, as opposed to the methods). Background: Citations are needed for the last two sentences in the third paragraph of the background describing benefits of co-production, or further explanation about why these benefits are thought to be attributable to co-production. Providing a specific example of co-production would be helpful, or of both types when mentioned in lines 24-36 of page 4. Methods: The methods are described very clearly and are rigorous. I recommend the authors refer to this as a systematic review, and make small updates to their manuscript to get credit for the work they have completed. AMSTAR criteria may be helpful in determining what might need to be added (e.g. a list citations for excluded studies), but it seems like the work has been done. The search, however, seems to be missing key terminology relating to patient centered work and patient engagement. While conducting a
---

	full search may be beyond the scope of this project, some discussion of why these terms were not included seems warranted, especially since they are very common terms for this type of work in the US (I admit I can't speak to other contexts). Results: Could the quality assessment rating criteria be incorporated as a supplementary file? There appear to be multiple checklists and more specificity would be helpful. In figure 1, the box with "full-text articles excluded, with reasons" is missing the reasons for the 13 exclusions. Discussion: While there is discussion of the limitations of the literature found, limitations for the review itself are not evident. A short paragraph describing any limitations of the review process used by the authors would remedy this. Do not hesitate to contact me with any questions.
--	---

VERSION 1 – AUTHOR RESPONSE

Reviewer 1:

-I would like to suggest some description of the co-produced interventions included in the papers. You identify significant variance in the types however the practical enactment is not included.

-Also some consideration into how this variance may influence outcomes could be discussed i.e. what is considered a co-produced intervention is far from standardized. We understand both of these points. We briefly review the issue of the meaning of co-production and how this has been defined on pages 3 and 4. Summary information on co-produced interventions reported for each study has been included in Supplementary Information file 2. Based on this feedback we have also expanded the discussion on pages 7-8 to respond to your suggestions.

Reviewer 2:

- I do think that there might be some work that uses 'co-production' but doesn't recognise or describe it in such a way (the Clinical Microsystems work from Dartmouth) which would not have appeared in your search, but would have had more of a focus on measurement of patient or service outcomes. We agree that this may be the case, we did not include or combine the terms 'clinical microsystems' in our search but we are aware of the work of Paul Batalden and colleagues at the Dartmouth Institute. We have now commented on this as a limitation of our search strategy (abstract and page 10) and acknowledged the clinical microsystems literature (see pages 10-11).

- On a very minor note the US spellings of recognize and other 'izes' are used throughout the document .

We acknowledge this point, but our understanding is that the journal house style is to use US spellings and so we have not made any changes in this regard.

Reviewer 3:

-p6, lines 20-21. Clarify how decisions on full text papers were made. Should retained full text papers" be "retrieved full text papers"? Did all three reviews independently make decisions on all retrieved papers? How were final decisions on inclusions arrived at?

-p6, lines 23-24. A bit more information on quality assessment would be helpful. It is not clear from the text what is included in the checklists and why they were selected (the relevant checklists could also be added as supplementary material).

-p6, line 56. Should (n=4) be (n=1)?

- p7, Reported outcomes of co-produced interventions. More information on any difference in how outcomes were measured would be helpful (e.g. methods for obtaining patients and carers views)

-p8, line 44. Typo - missing "to" (...required them to engage...)

-p8, line 49. Typo - missing "]" on reference 34.

-p22, Prisma diagram. The number of records excluded at Stage 1 doesn't add up, given the number of records screened and the number of full text articles assessed.

Thank you for these comments and suggestions.

We have amended the paragraph on page 6 to clarify the process we adopted.

We have added a section to paragraph 2 on page 6. This explains our rationale for the selection of the checklists developed by NICE and provides a brief summary of the nature of the checklists. We have now included a copy of the checklists list in a supplementary file (3).

We have amended line 56 on page 6 to read: A further study evaluated co-design projects in five Dutch hospitals, the projects were conducted in four oncology departments and a haematology department.

The introduction to the section on page 7 has been amended to outline the methods used to understand participants' experiences and obtain their views. The information in supplementary file 2 reporting on these methods is also highlighted.

Page 8 typos have been amended.

Page 22-Thank you for noting this error. We have amended this, the corrected version shows 688 records excluded at stage 1.

Reviewer 4:

In general, this manuscript could benefit from shortening and major editing to improve writing clarity - primarily in the form of reducing jargon. For example, in the Conclusion paragraph on page 11, there is the phrase, "At the clinical microsystem level within the healthcare sector..." If this means 'in acute health care clinics', the readers would benefit from shortening the verbiage as such. The following are some additional specific suggestions for refining the manuscript:

1. Abstract - Findings: Suggest revising to more completely summarizing findings. For example, outcomes evaluated are listed, but the actual results of the RCT seem to have been completely omitted. Suggesting listing a few of the major barriers and facilitators. Also, figure indicated 712 articles were screened rather than 710.

2. Strengths and limitations box on page 3: Suggest elaborating on these in the Discussion

3. Background - last sentence on page 3 is a run-on and should be split into two sentences.

4. Background - 3rd paragraph on page 4: Strongly suggest reducing the jargon. I read this paragraph numerous times and still found it difficult to understand.

5. Last sentence in first paragraph that cites evidence that policy makers increasingly value RES': Suggest adding recent relevant publication of using RES in a health care system: Peterson, K., Floyd, N., Ferguson, L., Christensen, V., & Helfand, M. (2016). User survey finds rapid evidence reviews increased uptake of evidence by Veterans Health Administration leadership to inform fast-paced health-system decision-making. *Systematic Reviews*, 5(1), 132.

6. Page 5 - 2nd paragraph: Limiting the scope to a specific focus would not widely be considered a "bias"; rather, this would limit applicability to wider audiences. Suggest rephrasing and adding additional biases common to RES such as widely discussed in the RES literature; e.g., limited critical appraisal, etc.

7. 'Elements of co-production that led to quality improvement' paragraph on page 8: Particularly for this paragraph - but elsewhere as well - suggest stating bottom line at beginning of section. I re-read this section a few times and was not able to decipher the main findings. The 'Economic Evaluation' section on the following page is a good example of starting the section with a clearly stated bottom line up front.

8. 'Barriers and facilitators to implementation' on page 9: 2nd paragraph lists a bunch of barriers.

Then, the subsequent sentences seem to be describing specific examples of barriers. But, it is difficult to link the examples to the preceding categories listed. What are 'practical or logistical' or 'recruiting' barriers examples of from the preceding list (i.e., "lack of support, resources or managerial authority to bring about structure or environmental changes").

9. Table 1 - inclusion criteria on page 5: Outcomes listed are very vague. Suggest providing some specific examples for context.

10. Discussion could also benefit from more detail on clinical and research implications. Authors state that their RES findings helped them in planning their CREATE study by indicating key areas for their research team to address. But, that is very vague. Suggest providing at least one specific example for context. What key practical messages should others interested in co-producing interventions take away from your review?

11. Future research: Authors suggest that future studies should focus on clinical and cost-effectiveness - but how? Suggest providing some more specific suggestions in the Discussion. For example, on page 7, authors emphasize the lack of outcome measurement using validated tools. In the Discussion, suggest revisiting this issue by proposing not just more research, but more robust research in terms of what key outcomes to measure and how they should be measured. What is the bar that a researcher to strive to reach?

We thank the reviewer for the range of comments and suggestions provided. We have had to strike a balance in terms of responding to the mainly positive comments of five different reviewers. In response to these specific comments we have not shortened the manuscript but we have taken account of the need to reduce jargon. The term clinical microsystem doesn't actually refer to acute healthcare clinics/hospitals but rather to spaces where patients, families, and care teams meet; these change over time and always have a patient or person with a health need at their centre. We have provided a definition and citation (Nelson et al, 2011) so as to make our meaning clearer.

1. We note these comments. We could not add examples of the key barriers and facilitators identified to the abstract without exceeding the word limit allowed in the submission process. In terms of the RCT results, we have not listed study specific findings for the other ten studies but rather summarised these in the abstract. We believe that the RCT findings in terms of co-design/co-production outcomes are sufficiently represented in the summary findings listed as a-c. The number of articles screened has been corrected to read 712. Thank you for spotting this error.

2. We have now made specific reference to these in the discussion section.

3. We have amended the last sentence on page 3 (commencing The evidence base in terms of robust evaluation) as suggested.

4. We note this comment. We have reviewed this paragraph again as a group and, whilst we acknowledge your perspective, we feel the wording is consistent with the argument developed in the preceding paragraphs and so have not revised the paragraph.

5. This is a helpful addition; we have added the reference as suggested, thank you.

6. The text in this paragraph on page 5 has been revised to take account of this comment.

7. We appreciate the view expressed here in terms of writing style. We have looked again at this and other paragraphs and made revisions to the text in order to make our findings clearer where that seemed appropriate.

8. We can understand how this may have read. The examples of barriers which follow from line 5 of this paragraph are not intended as examples (of the three barriers reported on line 4) but rather as a range of (often) linked reported barriers. We have amended the text slightly in this paragraph to make this clearer.

9. We note this comment. During the review process the team agreed to keep the outcomes deliberately broad in order to be as inclusive as possible of what we anticipated would be a limited evidence base. Adding examples as suggested would not reflect the actual process in terms of applying inclusion and exclusion criteria that we used in the review.

10. The discussion section has been revised to address both this comment and the comment below,

please see pages 11-12 of the revised manuscript.

11. Please see above.

Reviewer 5:

1. Abstract: There are a few grammatical issues (e.g., line 35 says "focus on clinical" but may need another word to clarify, clinical impact? clinical outcomes?). Both in the abstract and full text, articles are categorized as a feasibility RCT, 3 process evaluations, and 7 qualitative methods papers. This isn't quite parallel, since qualitative methods are not a study design. It would be more appropriate to say "descriptive qualitative" if these are not evaluations and only provide descriptive information (or describe the appropriate design, as opposed to the methods).
2. Background: Citations are needed for the last two sentences in the third paragraph of the background describing benefits of co-production, or further explanation about why these benefits are thought to be attributable to co-production. Providing a specific example of co-production would be helpful, or of both types when mentioned in lines 24-36 of page 4.
3. Methods: The methods are described very clearly and are rigorous. I recommend the authors refer to this as a systematic review, and make small updates to their manuscript to get credit for the work they have completed. AMSTAR criteria may be helpful in determining what might need to be added (e.g. a list citations for excluded studies), but it seems like the work has been done. The search, however, seems to be missing key terminology relating to patient centered work and patient engagement. While conducting a full search may be beyond the scope of this project, some discussion of why these terms were not included seems warranted, especially since they are very common terms for this type of work in the US (I admit I can't speak to other contexts).
4. Results: Could the quality assessment rating criteria be incorporated as a supplementary file? There appear to be multiple checklists and more specificity would be helpful. In figure 1, the box with "full-text articles excluded, with reasons" is missing the reasons for the 13 exclusions.
5. Discussion: While there is discussion of the limitations of the literature found, limitations for the review itself are not evident. A short paragraph describing any limitations of the review process used by the authors would remedy this.

Thank you for these comments.

1. We have amended this sentence in the abstract to make the meaning more clear. We note and accept the comment regarding qualitative methods. One of the challenges in the evidence synthesis was that few publications reported the study design adopted. In data extraction we asked reviewers to comment on both design and data collection methods. We have amended the abstract and full text to reflect our attribution of study design classification and made reference to commonly employed data collection methods.
2. Citations have been added
3. Thank you for this feedback. We have retained the rapid evidence synthesis title for the work undertaken but have added the terms systematic review to the key words. In preparing the manuscript we utilised the PRISMA checklist and included a copy of this (as recommended by BMJ Open) with the submitted manuscript. We did not think using the AMSTAR checklist would add value to the manuscript but we acknowledge the use of this tool in systematic review quality appraisal.
4. We acknowledge the comment regarding the omission of some commonly used search terms relating to patient centred work. We have commented in more detail on this in paragraph 3 on page 5. As you suggest this was a planned and pragmatic decision related to keeping the scope of the RES narrowly on co-production and co-design. We have now added this limitation in the discussion section of the paper and in the 'Strengths and Limitations' box on page 3 of the revised manuscript.
5. We have added a supplementary file (#3) which provides an example of the full quality assessment criteria used. We have also explained where the criteria originated and why we used these particular checklists on page 5 of the manuscript. The reasons for exclusion have now been listed in the PRISMA diagram. We have now added a short paragraph addressing the limitations of the review as suggested.

VERSION 2 – REVIEW

REVIEWER	Gemma Stacey University of Nottingham UK
REVIEW RETURNED	20-Feb-2017

GENERAL COMMENTS	Thank you for your detailed response to the reviewers comments. I am fully satisfied with the responses.
--

REVIEWER	Mr Daniel Wolstenholme NIHR CLAHRC YH, United Kingdom
REVIEW RETURNED	24-Feb-2017

GENERAL COMMENTS	Thanks for a comprehensive response to the reviewers comments. Look forward to seeing the article in print.
---

REVIEWER	Alison Eastwood CRD, University of York, UK
REVIEW RETURNED	13-Mar-2017

GENERAL COMMENTS	Thank you for sending me the revised manuscript to review. The authors have addressed the comments and amended the manuscript accordingly. There is just one point I would make regarding the response to my comment about quality assessment. The authors have helpfully provided more explanation in the text, however the details of their quality assessment is still not available to the reader - only the final overall rating is provided in Table 2. My suggestion would be to include the completed checklists for all the included studies as supplementary file 3, rather than only the blank qualitative checklist which is already clearly referenced in the text (reference 28).
---

REVIEWER	Kim Peterson Evidence-based Synthesis Program Coordinating Center, VA Portland Health Care System, USA
REVIEW RETURNED	28-Feb-2017

GENERAL COMMENTS	Thank you for the opportunity to review this revised manuscript. I commend the authors for their speedy resubmission. After reviewing the peer reviewers' comments, the author responses, and the revised manuscript, I conclude that the authors' changes have mostly adequately addressed the peer reviewers' suggestions to add important details and improve the overall clarity of the manuscript. The one exception is regarding the search. I agree with reviewers #2 and #5 that the search seems to be missing key terminology; e.g., clinical microsystems, patient-centered research and patient engagement. I appreciate that the authors made this decision for pragmatic reasons to keep the scope of the RES narrow and have added this as a limitation to the Discussion section. However, I suggest that the authors should also add more information to the Discussion about this limitation to better explain to the readers the
--

	potential impact of this decision and how it may or may not have affected their findings. I think this could be simply remedied by conducting an expanded search and screening the additional citations and commenting on whether or not this could have identified additional studies and how those might or might not differ from the authors' set of included studies. As the authors' search is already quite broad and already includes some terms for human-centered, people-centered, and user-centered, it may not actually add many citations and could be resolved as a non-issue? Another option could be to comment on how studies that use the terminology "co-production" or "co-design" are known to be fundamentally different than those that actually actively involved stakeholders to design and/or produce interventions, but didn't use that terminology and instead used the terminology of "engagement" or "centered," which I agree is quite common in the US. Maybe such studies would be less likely to use the formal forms of co-production of interest, such as AEBCD, EBD, Rotman, EBCD, etc., that are of the highest priority to the authors? In any case, I think it is not enough to only state the limitation, without some information about potential impact. Also, the placement of the new text about this limitation as the second sentence in the Discussion seems awkward. I suggest moving down to follow the authors' summary of findings.
--	--

REVIEWER	Isomi Miake-Lye VA Greater Los Angeles
REVIEW RETURNED	28-Feb-2017

GENERAL COMMENTS	These revisions are responsive and thoughtful. Thank you!
---

VERSION 2 – AUTHOR RESPONSE

Reviewer 3:

There is just one point I would make regarding the response to my comment about quality assessment. The authors have helpfully provided more explanation in the text, however the details of their quality assessment is still not available to the reader - only the final overall rating is provided in Table 2. My suggestion would be to include the completed checklists for all the included studies as supplementary file 3, rather than only the blank qualitative checklist which is already clearly referenced in the text (reference 28).

Thank you for these comments and suggestions.

We have included the completed quality assessment documents as suggested in Supplementary file 3.

Reviewer 4:

Thank you for the opportunity to review this revised manuscript. I commend the authors for their speedy resubmission. After reviewing the peer reviewers' comments, the author responses, and the revised manuscript, I conclude that the authors' changes have mostly adequately addressed the peer reviewers' suggestions to add important details and improve the overall clarity of the manuscript.

The one exception is regarding the search. I agree with reviewers #2 and #5 that the search seems to be missing key terminology; e.g., clinical microsystems, patient-centered research and patient engagement. I appreciate that the authors made this decision for pragmatic reasons to keep the scope of the RES narrow and have added this as a limitation to the Discussion section. However, I

suggest that the authors should also add more information to the Discussion about this limitation to better explain to the readers the potential impact of this decision and how it may or may not have affected their findings. I think this could be simply remedied by conducting an expanded search and screening the additional citations and commenting on whether or not this could have identified additional studies and how those might or might not differ from the authors' set of included studies. As the authors' search is already quite broad and already includes some terms for human-centered, people-centered, and user-centered, it may not actually add many citations and could be resolved as a non-issue? Another option could be to comment on how studies that use the terminology "co-production" or "co-design" are known to be fundamentally different than those that actually actively involved stakeholders to design and/or produce interventions, but didn't use that terminology and instead used the terminology of "engagement" or "centered," which I agree is quite common in the US. Maybe such studies would be less likely to use the formal forms of co-production of interest, such as AEBCD, EBD, Rotman, EBCD, etc., that are of the highest priority to the authors? In any case, I think it is not enough to only state the limitation, without some information about potential impact. Also, the placement of the new text about this limitation as the second sentence in the Discussion seems awkward. I suggest moving down to follow the authors' summary of findings.

Thank you for your time on this and for your comments.

We have carefully considered the points raised here; these were discussed in the research team as well as in more general terms with the BMJ Open editor. As you will be aware we undertook the rapid evidence synthesis, and made (as you point out) pragmatic decisions about the scope of searches undertaken based on the timescale and resources available for such reviews. In our view, undertaking additional searches post-hoc, as suggested changes the nature of the review we designed and conducted. We understand why you have asked us to consider this but we have chosen not to take this option.

In terms of the other suggestions you made re developing and reordering our discussion of the limitations of our search strategy and the impact that may have had on the study findings, we agree this was necessary and we have addressed these issues on page 11 of the manuscript.

VERSION 3 – REVIEW

REVIEWER	Gemma Stacey University of Nottingham Uk
REVIEW RETURNED	04-Apr-2017

GENERAL COMMENTS	Additional feedback given by some reviewers has been carefully considered and addressed.
--

REVIEWER	Dan Wolstenholme Sheffield Teaching Hospitals NHS Foundation Trust, NIHR CLAHRC YH
REVIEW RETURNED	06-Apr-2017

GENERAL COMMENTS	thanks again for responding to the peer reviewers comments
--

REVIEWER	Alison Eastwood CRD, University of York, UK
REVIEW RETURNED	06-Apr-2017

GENERAL COMMENTS	The authors have addressed my suggestion and provided the completed quality assessment checklists as a supplementary file.
--

REVIEWER	Kim Peterson Evidence-based Synthesis Program (ESP) Coordinating Center, VA Portland Health Care System, Portland, OR, USA
REVIEW RETURNED	11-Apr-2017

GENERAL COMMENTS	Thanks to the authors for reordering and further developing their discussion of the potential impact of their search strategy on their review findings. I am satisfied with their changes.
--

REVIEWER	Isomi Miake-Lye VA Greater Los Angeles, USA
REVIEW RETURNED	17-Apr-2017

GENERAL COMMENTS	Agree with decisions made by authors, very responsive.
--